# Plasma Osteopontin Reflects Tissue Damage in Acute Pancreatitis

**DOI:** 10.3390/biomedicines11061627

**Published:** 2023-06-03

**Authors:** Lina Wirestam, Pernilla Benjaminsson Nyberg, Todor Dzhendov, Thomas Gasslander, Per Sandström, Christopher Sjöwall, Bergthor Björnsson

**Affiliations:** 1Department of Biomedical and Clinical Sciences, Division of Inflammation & Infection, Linköping University, SE-581 85 Linköping, Sweden; lina.wirestam@liu.se; 2Department of Surgery in Linköping, Linköping University, SE-581 83 Linköping, Sweden; pernilla.benjaminsson.nyberg@regionostergotland.se (P.B.N.); todor.dzhendov@regionostergotland.se (T.D.); thomas.gasslander@liu.se (T.G.); per.sandstrom@liu.se (P.S.); bergthor.bjornsson@regionostergotland.se (B.B.); 3Department of Biomedical and Clinical Sciences, Division of Surgery, Orthopedics and Oncology, Linköping University, SE-581 85 Linköping, Sweden

**Keywords:** osteopontin, acute pancreatitis, organ damage, biomarker

## Abstract

Several scoring systems for clinical prediction of the severity of acute pancreatitis (AP) have been proposed. Yet, there is still a need for an easy-to-measure biomarker. Osteopontin (OPN) may be released to the circulation early during tissue injury, but the significance of OPN in AP has not yet been established. We aimed to evaluate plasma levels of OPN in relation to the severity of AP. In 39 individuals with confirmed AP, plasma was collected on the day of admission and consecutively for three days thereafter. Sex- and age-matched healthy blood donors (*n* = 39) served as controls. Plasma OPN was measured by a commercial enzyme-linked immunosorbent assay. At admission, patients with AP displayed higher OPN, 156.4 ng/mL (IQR 111.8–196.2) compared to controls, 37.4 ng/mL (IQR 11.7–65.7) (*p* < 0.0001). However, OPN levels on admission could not discriminate between mild and moderate-to-severe disease (132.6 ng/mL vs. 163.4 ng/mL). Nevertheless, the changes in OPN within 24 h of admission and Day 2/3 were higher among patients with moderate/severe AP (33.7%) compared to mild AP (−8.1%) (*p* = 0.01). This indicates that OPN is a relevant biomarker reflecting tissue injury in AP. The increase in OPN over time suggests that serial OPN measurements could contribute to the early detection of at-risk patients. Prospective studies assessing OPN in relation to outcome in AP are warranted.

## 1. Introduction

Acute pancreatitis (AP) is a condition with considerable variation in severity that can be hard to predict early at disease onset [1]. The majority, approximately 80%, develop mild (no organ failure or local complications) to moderately severe disease (local complications and/or organ failure < 48 h). The remaining fifth develop severe disease (organ failure > 48 h), which has a mortality rate of 20%. Different clinical scoring systems, such as the Acute Physiology and Chronic Health Evaluation (APACHE) II score, as well as various biomarkers for the prediction of organ failure and severity of AP, have been proposed over the years [2,3]. Nowadays, for AP and many other diseases, C-reactive protein (CRP) is the gold standard biomarker for risk stratification in clinical routine [1,2,4]. However, there is still an urgent need for more precise biomarkers.

Osteopontin (OPN), also known as bone/sialoprotein I, early T-lymphocyte activation-1, and secreted phosphoprotein 1, is a bone matricellular protein that is involved in bone turnover but is also released by inflammatory and other cell types when there is tissue injury [5,6,7]. It is involved in both cell–cell and cell–matrix interactions crucial in the inflammatory response and can act as a soluble cytokine in inflamed tissues and circulation [8]. OPN is widely distributed in normal adult human tissues, being abundantly expressed by osteoclasts and osteoblasts in bone matrix and present in kidneys, epithelial cells of the gastrointestinal tract, gall bladder, pancreas, urinary and reproductive tracts, lungs, breast, salivary glands, sweat glands, inner ear, brain, decidua, placenta, arteries, human blood, and body fluids such as urine and milk [9].

Prior studies have shown that serum OPN levels are high in systemic inflammatory response syndrome (SIRS) and severe sepsis/septic shock and correlate with levels of interleukin 6 (IL-6) [6]. As recently reviewed by Frittoli et al., pancreatic involvement can also be found in patients with autoimmune inflammatory diseases, such as systemic lupus erythematosus (SLE), and present with symptoms such as acute abdominal pain, fever, nausea, and vomiting [10]. Of relevance in this context, circulating OPN was shown to correlate with disease activity in individuals with recent-onset SLE and associated with secondary antiphospholipid syndrome [11]. In AP, OPN levels on admission have been shown to correlate with APACHE II scores as well as IL-6 levels [12]. Moreover, OPN is associated with increasing pancreatitis severity [12] and could be a potential early marker of mortality in AP patients [13].

As OPN has been proposed as a relevant biomarker reflecting or even preceding tissue injury, we herein aimed to evaluate plasma levels of OPN in relation to the severity of AP.

## 2. Materials and Methods

### 2.1. Study Subjects

This was a prospective study including 39 patients with AP (mean age 48.5, range 21–70 years, 54% women) admitted to the Department of Surgery, University Hospital of Linköping, Sweden, between 2011 and 2016. Baseline characteristics of patients included in this study are demonstrated in Table 1. The diagnosis of AP was confirmed according to the 2012 Atlanta classification in the emergency department when at least two out of three of the following criteria were positive: typical clinical presentation of AP with acute abdominal pain, serum amylase levels at least three times higher than the upper limit of normal and typical findings of AP on contrast-enhanced computed tomography [14]. In addition, clinical data were collected (age, sex, etiology, body mass index, onset of symptoms, length of hospitalization, admission to intensive care unit, SIRS, and need for rehospitalization within 90 days). Systemic complications and the severity of AP were determined according to the Atlanta classification [14]. Data on AP severity based on radiology, multiple organ dysfunction, and comorbidities were assessed by the Balthazar score [15], Marshall score [16], and Charlson comorbidity index [17], respectively.

Blood samples were collected on the day of admission to the hospital (Day 0) and then on consecutive days until Day 3 for those patients still admitted to the ward. Thirty-nine professional blood donors, age and sex-matched to the AP study population (mean age 48, range 21–70 years, 51% women), served as controls and were recruited from the Department of Transfusion Medicine at University Hospital of Linköping.

### 2.2. Laboratory Analyses

Plasma samples, stored at −70 °C until analysis, were analyzed in duplicates for OPN by enzyme-linked immunosorbent assay (ELISA) according to the manufacturer’s instructions (Quantikine^TM^, Techne Corporation R&D Systems, Minneapolis, MN, USA) as previously described [11]. Briefly, plasma (diluted 1:25) was added to ELISA plates, precoated with monoclonal antibodies directed against human OPN. After incubation and washing of the wells, a horseradish peroxide conjugated polyclonal OPN specific antibody was added, and the plate was incubated, followed by washing and addition of tetramethylbenzidine substrate. The enzymatic reaction was stopped by adding 2 N sulfuric acid and read at 450 nm (plate reader Sunrise, Tecan, Männedorf, Switzerland; software Magellan V.7.1, Tecan).

IL-6 in plasma was assessed using a colorimetric immunoassay (detection limit 1.5 ng/L). Routine blood samples, including procalcitonin, amylase, and CRP, were analyzed at the Department of Clinical Chemistry at University Hospital of Linköping as part of clinical routine [18].

### 2.3. Statistics

Possible differences between two groups were analyzed using Mann–Whitney *U* test. Kruskal–Wallis test with Dunn’s multiple comparison tests was applied when analyzing three or more groups. Chi2 test was used for analyzing two dichotomous variables. Binary logistic regression was used to predict disease severity, with adjustment for age and sex. *p*-values below 0.05 were considered significant. Statistical analyses were performed with SPSS Statistics 29 (IBM, Armonk, NY, USA) or GraphPad Prism 9 (GraphPad Software, La Jolla, CA, USA).

## 3. Results

### 3.1. OPN Levels in AP and Controls

At admission, patients with AP displayed higher OPN, median 156.4 ng/mL (IQR 111.8–196.2) compared to the controls, 37.4 ng/mL (IQR 11.7–65.7), *p* < 0.0001 (Figure 1A). No significant differences were seen over time as indicated by the following results: Day 0, *n* = 19, median 155.5 ng/mL (IQR 101.0–188.0); Day 1, *n* = 23, median 168.3 ng/mL (IQR 125.1–251.7); Day 2, *n* = 25, median 173.2 ng/mL (IQR 133.2–244.7); and Day 3, *n* = 11, median 169.6 ng/mL (IQR 110.5–206.5) (Figure 1B).

The most common etiologies of AP were gallstone disease, alcohol, and idiopathic (*n* = 35, 90%). Plasma OPN did not differ significantly on admission, or remaining time points, between the etiologies.

### 3.2. OPN Levels and AP Severity

OPN levels on admission could not discriminate between mild (*n* = 21, 163.4 ng/mL) and moderate-to-severe disease (*n* = 9, 132.6 ng/mL).

Over time, patients with mild versus (vs.) moderate-to-severe disease displayed the following: Day 0, *n* = 14, median 164.7 ng/mL (IQR 104.3–206.2), vs. *n* = 5, median 124.7 ng/mL (IQR 98.8–156.2). Day 1; patients with mild disease, *n* = 16, median 164.5 ng/mL (IQR 116.8–222.4), vs. patients with moderate-to-severe disease, *n* = 7, median 251.7 ng/mL (IQR 129.8–437.5). Day 2; patients with mild disease, *n* = 16, median 140.6 ng/mL (IQR 132.5–175.7), vs. patients with moderate-to-severe disease, *n* = 9, median 265.9 ng/mL (IQR 164.2–348.8). Day 3; patients with mild disease, *n* = 7, median 158.7 ng/mL (IQR 94.7–189.8) vs. patients with moderate-to-severe disease, *n* = 4, median 214.3 ng/mL (IQR 138.2–330.4). A significant difference was only seen between patients with mild vs. moderate-to-severe disease on Day 2 (*p* = 0.005, Figure 2A).

Twenty-one patients had samples taken within 24 h of admission, and the last sample was taken on Day 2 or 3. No differences were seen between patients with mild AP within 24 h of admission (*n* = 14) vs. Day 2/3 (median 160.4 ng/mL, IQR 106.2–195.9 vs. 151.3 ng/mL, IQR 113.5–174.5). However, patients with moderate-to-severe disease (*n* = 7) displayed higher OPN levels at Day 2/3 vs. within 24 h of admission (median 222.2 ng/mL, IQR 139.8–395.4 vs. median 146.3 ng/mL, IQR 124.7–251.7). Hence, the percentage change between admission and Day 2/3 was higher among patients with moderate-to-severe AP (33.7%) compared to mild AP (−8.1%) (*p* = 0.01, Figure 2B).

None of the patients included died from their illness. Patients with signs of pulmonary edema displayed higher levels of OPN (median 318.3 ng/mL, IQR 199–437.5) compared to those without (median 148 ng/mL, IQR 98.3–195.9), but this difference did not meet statistical significance (*p* = 0.12). OPN levels within 24 h of admission did not correlate with APACHE II (rho = 0.41, *p* = 0.17). No correlation was found between OPN levels within 24 h of admission and amylase (rho = −0.06, *p* = 0.75).

Binary logistic regression was used to predict disease severity, with adjustment for age and sex. The disease severity predictive value for OPN did not reach statistical significance, nor did it when analyzing the possible predictive value of CRP, IL-6, and procalcitonin one by one. A combined model with OPN, CRP, IL-6, procalcitonin, age, and sex did not render any statistical significance.

## 4. Discussion

This prospective cohort study of patients admitted to the Department of Surgery, University Hospital of Linköping, adds further support for OPN as a potential early marker of tissue injury in AP. The main findings include a significant elevation of OPN on admission compared to matched healthy blood donors, and dynamic changes over time significantly related to disease severity. Furthermore, the prognostic value was found to be at least as reliable as the established biomarker of inflammation used in the clinic—CRP.

Regarding the prediction of clinical outcomes in other conditions, OPN has previously been evaluated in SLE [9]. Rullo et al. reported that increased circulating OPN levels preceded increased cumulative disease activity and global organ damage in patients with SLE, especially in pediatric SLE [19]. Less clear results were found in adult SLE where the potential of OPN to predict irreversible organ damage was studied in 344 patients with the recent-onset disease. Elevated levels of OPN did not significantly predict future organ damage but were shown to associate with renal involvement (lupus nephritis) and with raised disease activity at enrollment, as well as over time [20].

Studies evaluating OPN as a biomarker for AP are scarce, warranting additional investigations. Plasma OPN has earlier been reported to be associated with AP severity [12,13]. In our study, OPN levels within 24 h of admission could not discriminate between patients with mild/severe disease which is in line with the findings recently reported by Rao et al. [13]. In that study, similar OPN levels in patients with and without organ failure were observed, whereas a strong association with mortality was reported [13]. In contrast, none of the included patients in our study died. This significantly limits the possibility to make a fair comparison between the studies.

In a somewhat larger study by Swärd et al., including 86 patients, OPN levels were found to predict future organ failure [12]. Although OPN levels on admission could not differentiate between mild and moderate-to-severe disease in our cohort, significantly higher levels were seen in patients compared to controls, indicating inflammation and/or tissue injury. Moreover, a dynamic change was seen over time where patients with severe disease displayed higher levels, albeit only significant on Day 2. The relative change between admission and Day 2/3 was also higher among patients with moderate-to-severe disease.

Obviously, the time of sampling and length of symptoms are important when a biomarker is evaluated for the prediction of outcome. It is possible that this could explain the difference between our findings and those reported by Swärd et al. However, in line with the observation by Swärd et al., we observed no differences in OPN between the different AP etiologies [12].

OPN was significantly increased in AP patients on admission compared to healthy controls, indicating tissue injury. Another biomarker used in the diagnosis of AP is amylase, which rises rapidly three to six hours after the onset of symptoms to three times the upper limit of normal [21]. Amylase is a good biomarker for diagnosis; however, it lacks the ability to predict AP severity [21]. We observed no significant correlation between levels of OPN and amylase.

This study has limitations that need to be acknowledged. The number of included patients and controls was relatively low, resulting in restricted statistical power. Lack of difference in comparisons between groups should therefore be interpreted with caution. It would be of great value to investigate OPN in a larger cohort of patients whereof a greater proportion of patients would have severe AP to establish the predictive value of OPN. Moreover, further studies could result in a cut-off value to distinguish patients with mild from moderate-to-severe disease. In our cohort, patients with moderate-to-severe disease display OPN levels above 200 ng/mL from Day 1 and onwards. The use of OPN as a clinical biomarker would also require cheap and clinically validated kits. Another limitation is that this study was retrospectively registered at ClinicalTrials.gov.

## 5. Conclusions

Despite the limitations mentioned above, our results indicate that OPN is a promising biomarker reflecting tissue injury in AP. The increase in OPN over time suggests that serial OPN measurements could contribute to the early detection of at-risk patients. Nonetheless, the clinical value of OPN needs to be further evaluated in larger cohorts.

## Figures and Tables

**Figure 1 biomedicines-11-01627-f001:**
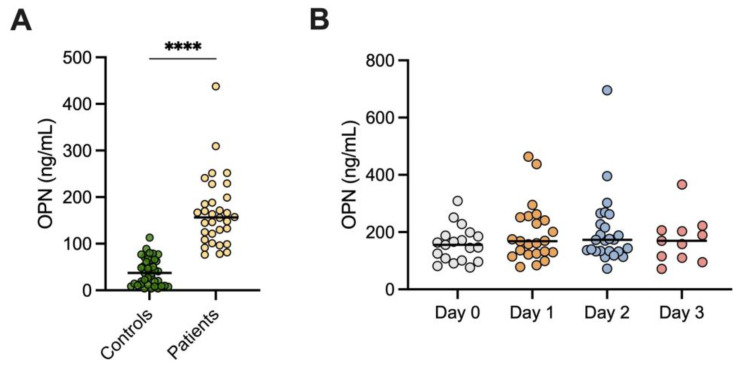
Osteopontin (OPN) levels in (**A**) patients with acute pancreatitis within 24 h of admission (AP) and healthy blood donors; and (**B**) patients with AP on the day of admission to the hospital (Day 0) and then consecutively until Day 3. **** *p* < 0.0001.

**Figure 2 biomedicines-11-01627-f002:**
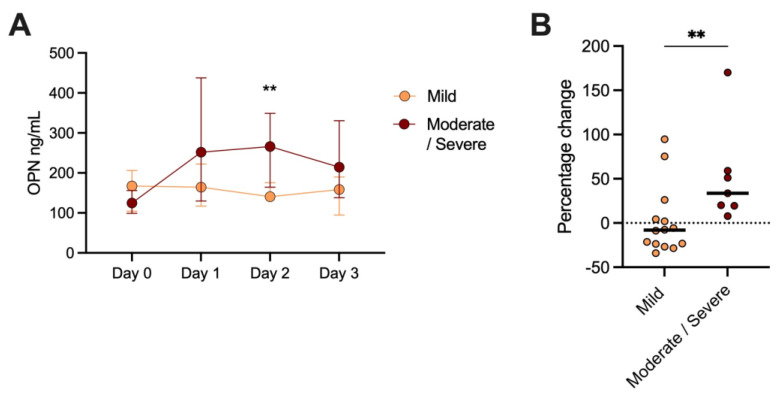
(**A**) Osteopontin (OPN) median (+/− IQR) levels over time in patients with mild and moderate-to-severe disease. (**B**) Percentage change in OPN levels between admission and Day 2/3 in patients with mild and moderate-to-severe disease. ** *p* < 0.01.

**Table 1 biomedicines-11-01627-t001:** Characteristics of patients with acute pancreatitis included in this study.

	All Patients	Mild (*n* = 26)	Moderate/Severe (*n* = 11)	
Sex, female, *n* (%)	20 (54)	14 (54)	6 (55)	ns
Age, median (IQR), years	54 (34–60)	54 (28–60)	56 (46–67)	ns
BMI, median (IQR)	28 (26–33.5)	28 (26–33)	29 (25–34.5)	ns
Prior AP, *n* (%)	10 (27)	9 (15)	1 (9)	ns
SIRS, *n* (%)	9 (24)	4 (15)	5 (46)	ns
CRP, median (IQR), mg/L	5 (5–19)	9 (5–27)	5 (5–7.5)	ns
IL-6, median (IQR), ng/L	45 (10–87)	35 (10–81)	55 (31–168)	ns
Procalcitonin (IQR), µg/L	0.13 (0.05–0.65)	0.2 (0.05–0.4)	0.05 (0.05–0.83)	ns
Amylase (IQR), µkat/L	18.3 (5.9–32.0)	20.5 (6.1–32.9)	10.1 (5.8–27.4)	ns
Length of hospital stay, median (IQR), days	5 (4–8)	5 (3–6)	10 (7.5–16.5)	<0.001
ICU care, *n* (%)	2 (5)	0 (0)	2 (18)	ns
Rehospitalization within 90 days, *n* (%)	4 (11)	2 (8)	2 (18)	ns
Charlson comorbidity index [17]	0 (0–1)	0 (0–0)	0 (0–1)	ns
Marshall score [16]	0 (0–0)	0 (0–0)	0 (0–0)	ns
Computed Tomography performed, *n* (%)	19 (51)	11 (42)	8 (73)	ns
Balthazar score [15]	2 (1–4)	2 (0–2)	4 (2–6)	ns

AP = acute pancreatitis, BMI = body mass index, CRP = C-reactive protein, ICU = intensive care unit, IL = interleukin, IQR = interquartile range, ns = not significant, SIRS = systemic inflammatory response syndrome.

## Data Availability

The data presented in this study are available on request from the corresponding author.

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
