# Peer review of "Plasma Osteopontin Reflects Tissue Damage in Acute Pancreatitis"

_biomedicines, 2023, doi:10.3390/biomedicines11061627_

Round 1

Reviewer 1 Report

This prospective study evaluated the utility of OPN as acute pancreatitis (AP) predictive biomarker. Although OPN could not show a significant difference between AP patients and donors, OPN was elevated in moretare-sever AP. I have some questions about this biomarker, including the comparison of the standard biomarker (Amy).

#1 It is difficult to understand the difference in the OPN volume between AP patients and donors. Please add the figure or table which explains the relation of the OPN volume between AP patients and donors.

#2 The Amy level is a well-known biomarker for AP, and this study also uses the Amy level for diagnosing AP. Please add the Amy level in Table 1.

For considering the utility of OPN, please add the data on the relationship between OPN and Amy. A discussion of the difference between OPN and Amy also require in the discussion part. 

#3 This study reported there was no significant difference in OPN volume between AP patients and donors. However, the moderate-sever AP patient had significantly higher OPN volume than the mild AP.

When used clinically, is it a good predictor of moderate or severe AP? If it may be possible, please consider the cut-off value of OPN for moderate or severe AP and discuss this point.

#4 Authors mentioned that OPN can elevate due to various inflammation. Moderate-severe AP may develop respiratory and renal failure, and an increase in OPN due to these may be expected.

Please add the presence or absence of organ failure, 90-day mortality, and its details to Table 1.

Author Response

Comments and Suggestions for Authors

This prospective study evaluated the utility of OPN as acute pancreatitis (AP) predictive biomarker. Although OPN could not show a significant difference between AP patients and donors, OPN was elevated in moretare-sever AP. I have some questions about this biomarker, including the comparison of the standard biomarker (Amy).

#1 It is difficult to understand the difference in the OPN volume between AP patients and donors. Please add the figure or table which explains the relation of the OPN volume between AP patients and donors.

Response: We believe that this could be a misunderstanding. In fact, the OPN concentrations in patients and donors were shown in Figure 1A illustrating 39 age- and sex-matched professional blood donors serving as controls.

#2 The Amy level is a well-known biomarker for AP, and this study also uses the Amy level for diagnosing AP. Please add the Amy level in Table 1.

Response: This is a relevant request. We have now added the requested data regarding amylase levels in Table 1 according to the reviewer’s suggestion.

For considering the utility of OPN, please add the data on the relationship between OPN and Amy. A discussion of the difference between OPN and Amy also require in the discussion part.

Response: We observed no significant correlation between OPN and amylase concentrations at admission (rho=–0.06, p=0.75). This information has now been added to the Result section (page 5, line 156-157). Amylase is an established biomarker for AP diagnosis, but it lacks the ability to predict severity in AP (see PMID 28720341). We have briefly discussed the difference between OPN and amylase (page 6, line 207-211).

#3 This study reported there was no significant difference in OPN volume between AP patients and donors. However, the moderate-sever AP patient had significantly higher OPN volume than the mild AP.

Response: In fact, we found significantly higher OPN levels in patients at admission (median 156.4 ng/mL, IQR 111.8–196.2) compared to the controls, (median 37.4 ng/mL, IQR 11.7–65.7) p<0.0001 which is shown in Figure 1A. This is stated first in the result section; 3.1 OPN levels in AP and controls.

When used clinically, is it a good predictor of moderate or severe AP? If it may be possible, please consider the cut-off value of OPN for moderate or severe AP and discuss this point.

Response: Future studies in larger cohorts investigating OPN could result in a cut-off value to distinguish patients with mild disease from those with moderate-to-severe disease. In our cohort patients with moderate-to-severe disease display OPN levels above 200 ng/mL from day 1 and onwards. This was added to the Discussion (page 6, line 218-220).

#4 Authors mentioned that OPN can elevate due to various inflammation. Moderate-severe AP may develop respiratory and renal failure, and an increase in OPN due to these may be expected.

Response: It is a relevant comment that OPN may increase in these conditions. Patients with signs of pulmonary edema displayed higher levels of OPN (median 318.3 ng/mL, IQR 199–437.5) compared to those without (median 148 ng/mL, IQR 98.3–195.9), but this difference did not meet statistical significance (p=0.12). We have also added data to Table 1 regarding patients needing intensive care unit (ICU). Unfortunately, we did not have access to data concerning renal failure.

Please add the presence or absence of organ failure, 90-day mortality, and its details to Table 1.

Response: In the Introduction of the previous manuscript version, we incorrectly mentioned that we had data available on 90-day mortality. This was a mistake. However, as a reflection of AP severity, we have added data regarding the proportion of patients needing ICU care as well as those regarding the need of rehospitalization within 90 days (see revised Table 1). 

Reviewer 2 Report

Wirestam et al. present a prospective study investigating Plasma osteopontin as a prognostic factor in acute pancreatitis. The rationale to perform this study is given. The methods are adequate and the reesults are well presented. The discussion is balanced.

I have some concerns to be addressed:

According to the Declaration of Helsinki, every prospective study in humans should be registered prior to the inclusion of the first patient. Please report the trial registry number. If this trial was not registered, please do so retrospectively and discuss it as a limitation.

Please add to the discussion what is the future reasech plan to confirm results and plasma osteopontin as a standard marker in acute pancreatitis. Please also discuss if there are any practical issues in daily clinical application (costs etc.).

Author Response

Comments and Suggestions for Authors

Wirestam et al. present a prospective study investigating Plasma osteopontin as a prognostic factor in acute pancreatitis. The rationale to perform this study is given. The methods are adequate and the reesults are well presented. The discussion is balanced.

I have some concerns to be addressed:

According to the Declaration of Helsinki, every prospective study in humans should be registered prior to the inclusion of the first patient. Please report the trial registry number. If this trial was not registered, please do so retrospectively and discuss it as a limitation.

Response: Thank you for this relevant request. Retrospectively, we have now registered our study at clinicaltrials.gov (see attached "ClinicalTrials.gov Protocol Registration and Results System Receipt"). We acknowledge this as a limitation in the Discussion (page 6, line 220-221). However, we still have not received the NCT number (page 7, line 244) but we plan to add it as soon as possible.

Please add to the discussion what is the future research plan to confirm results and plasma osteopontin as a standard marker in acute pancreatitis. Please also discuss if there are any practical issues in daily clinical application (costs etc.).

Response: It would be of great value to investigate OPN in a larger cohort of AP patients, whereof a larger proportion of patients develop severe AP. To our knowledge, the kits available are not yet clinically validated and are still rather expensive. We have now added this to the Discussion (page 6, line 218-220).

Round 2

Reviewer 1 Report

Thank you.

Thank you.

Author Response

Thank you for the careful review of our manuscript.

Reviewer 2 Report

Please add the NCT register number as soon as you receive it.

Author Response

Thank you for the careful review of our manuscript. The NCT register number has been added in the final version of the manuscript.